# Hidden Pitfalls of Using Onion Pollen in Molecular Research

**Majd Mardini** [1,†] **, Aleksey Ermolaev** [1,2,†] **and Ludmila Khrustaleva** [1,3,4,*]

1    Center of Molecular Biotechnology, Russian State Agrarian University—Moscow Timiryazev Agricultural Academy (RSAU-MTAA), 49, Timiryazevskaya Str., 127550 Moscow, Russia

2    Laboratory of Applied Genomics and Crop Breeding, All-Russia Research Institute of Agricultural Biotechnology, Timiryazevskaya Str. 42, 127550 Moscow, Russia

3    Department of Botany, Breeding and Seed Production of Garden Plants, Russian State Agrarian University—Moscow Timiryazev Agricultural Academy (RSAU-MTAA), 49, Timiryazevskaya Str., 127550 Moscow, Russia

4    Plant Cell Engineering Laboratory, All-Russian Research Institute of Agricultural Biotechnology, Timiryazevskay 42 Str., 127550 Moscow, Russia

*    Correspondence: ludmila.khrustaleva19@gmail.com or khrustaleva@rgau-msha.ru

†    These authors contributed equally to this work.

**Abstract:** There is little information on the use of pollen in molecular research, despite the increased interest in genome editing by pollen-mediated transformation. This paper presents an essential toolbox of technical procedures and observations for molecular studies on onion (*Allium cepa* L.) pollen. PCR is a useful tool as an express method to evaluate editing results before pollination. A direct PCR protocol for pollen suspension has been adapted without needing DNA pre-extraction. We showed that the outer layer of lipids known as pollenkitt is a limiting factor for successful PCR on pollen. A simple pre-washing step of pollen suspension was able to eliminate the pollenkitt and enormously affect the PCR results. Additionally, our pollenkitt study helped us develop a simple and effective pollination method using wetted onion pollen grains. Classical manual pollination usually is conducted by intact pollen without wetting. Most existing methods of the editing system delivery into pollen are carried out in a wet medium with consequent drying before pollination, which adversely affects the viability of pollen. The optimal medium for wet pollination was 12% sucrose water solution. Our method of using wetted pollen grains for pollination might be very beneficial for pollen genetic manipulation.

**Keywords:** onion; PCR inhibition; pollination; pollenkitt; wetted pollen

## 1. Introduction

For many years, pollen has been an appealing subject in a wide spectrum of studies, such as paleontology [1–3], forensics [4], vaccine development [5,6], allergology [7] and many others. Furthermore, pollen research is also considered a major player in the field of plant breeding and crop improvement. The fact that a pollen grain consists of 2–3 haploid cells gives a unique chance to conduct fundamental studies like DNA genotyping and population genetics [8–14]. Another promising line of research is the use of pollen as a natural carrier of exogenous DNA for genome editing [15–23]. Although many articles have been published on the study of pollen, there is scarce information on the use of pollen in molecular research.

PCR is the most used instrument in molecular biology for the analysis of DNA variations. In the case of genetic manipulation of pollen, PCR might be used as an express method to evaluate results before pollination. Since the genetic material inside pollen grains is well protected by the rigid wall, DNA extraction is considered a critical step for successful PCR. This step is usually conducted either by mechanical crushing or enzymatic lysis or a combination of the two [10,12,18]. However, some observations suggest that DNA might be passively released from pollen grains due to the high temperature during PCR

amplification [24], or in some certain species, pollen grains simply burst upon contact with liquid [25]. Another issue is the possible inhibition of PCR due to the natural presence of certain substances in pollen grains. A pre-washing step before PCR is obviously required in many protocols [3]; however, several studies claim that constituents of pollen grains have no effect on PCR activity [12]. Nevertheless, PCR on pollen might be a species-specific procedure. In addition, there is a marked lack of research on commercially important crops, highlighting the need to adapt protocols for new species.

Pollen-mediated genetic manipulation is a revolutionary approach in crop improvement. This promising technique gives the ability to readily obtain transformed plants in the form of seeds, which means bypassing the laborious steps of tissue culture and regeneration. Since the mid-1970s, scientists have been suggesting a wide range of methods to solve this problem, including free uptake of DNA [16], particle bombardment [19,22], electroporation of pollen tubes [17,18], pollen magnetofection [21], and uptake of nanoassemblies [23]. Regardless of the many distinctive variations among these methods, all of them require a *wet medium* to facilitate the main procedure. However, in most available protocols, little attention has been paid to the impact of wet mediums on pollen viability and subsequent pollination.

This paper presents an essential toolbox of technical procedures and observations for molecular studies on onion (*Allium cepa* L.) pollen. A direct PCR protocol for pollen suspension has been adapted without needing DNA pre-extraction. We showed that a limiting factor for successful PCR on pollen suspension is a material known as pollenkitt. Pollenkitt is characteristic of many entomophilous species but lacking in those which are wind pollinated. Pollenkit was first named by Knoll in 1930 and described as all substances which connect single pollen grains to pollen clumps [26]. Pollenkitt is a hydrophobic mixture, composed mainly of saturated and unsaturated lipids and also carotenoids, flavonoids, proteins, and carbohydrates [27]. The physico-chemical properties of pollenkitt depend on the substances in the mixture and their proportions [28]. Pollenkitt is present in large quantities in exine cavities of pollen [26]. A simple pre-washing step of pollen suspension was able to eliminate the pollenkitt and enormously affect the PCR results.

Additionally, phase-contrast and dark-field microscopy were used to determine the dynamics of pollenkitt release from pollen grains upon contact with a liquid medium. It was clear that pollenkitt plays a major role in increasing the surface area of the pollen grain, hence affecting floating and adhesion behavior. Our investigation on pollenkitt helped us to develop a simple and efficient method for pollination of onion flowers using wetted pollen grains. All the observations and methods in this paper will be useful for future research on gene editing and manipulation of onion pollen.

## 2. Materials and Methods

### 2.1. Plant Materials and Pollen Collection

Bulbs of onion (*Allium cepa* L.) cultivar "Myachkovsky 300" were kindly provided by the Federal Scientific Vegetable Center. By the end of April 2022, vernalized onion bulbs were planted within a 40 cm distance in an open field at The Center of Molecular Biotechnology—Russian State Agrarian University, Moscow. Flowering starts by the end of July 2022. Since flowering within one umbel is not synchronized, pollen was collected once or twice a day for 3–4 days. The pollen was used immediately or stored in a small Petri dish at −20 °C for further use. Mature seeds were collected by the end of August 2022.

### 2.2. PCR on Pollen Suspension

Using a sensitive analytical balance (Ohaus-PA214C), 0.5 mg of pollen was weighed inside a 1.5 mL microtube. A simple washing technique was performed by vortexing in 500 µL of distilled water for 10 s and centrifuged for 5 s using minispin centrifuge ($750 \times g$). After discarding the supernatant, fresh distilled water was added and the same procedure was repeated for 10 rounds. To ensure that no precipitated pollen grains are lost during washing, only 450 µL of the supernatant was carefully discarded between rounds. As a

result, after discarding the last supernatant, 50 μL of a pollen grains suspension with a concentration of 10 μg/μL was obtained.

PCR was performed using 10 μg of mature intact pollen (1 μL of the pre-washed suspension), or 50 ng of *A. cepa* genomic DNA as a positive control. Genomic DNA was isolated from young leaves by a standard CTAB-method [29]. 25 μL of $1\times$ PCR mix contained 2.5 μL of $10\times$ Taq Turbo buffer (25 mM $MgCl_2$, pH = 8.3) (Evrogen, Moscow, Russia), 0.2 mM of each dNTP (Evrogen, Moscow), 0.2 μM both forward and reverse primers and 5 U of Taq-polymerase (Evrogen, Moscow). Primers (F: TCAAGCACTGCAAACCTCTTC; R: ATGGAGCTAAATCCTGGTGCA) were designed for amplification of ORF (Open Reading Frame) of the LFS gene (GenBank: AB089203.1). Touchdown PCR was performed to amplify desirable amplicons without a non-specific PCR product. Cycling was performed as follows: 10 min of initial denaturation at 95 °C, 10 cycles (touchdown stage)—30 s of denaturation at 95 °C, 30 s of annealing (from 67 °C to 62 °C, decrement at 0.5 °C/cycle), 45 s of elongation at 72 °C, 30 cycles (standard PCR)—30 s of denaturation at 95 °C, 30 s of annealing at 62 °C, 45 s of elongation at 72 °C, 10 min of final elongation at 72 °C. PCR product was visualized in agarose gel (2% of agarose in $0.5\times$ TBE buffer, 0.5 μg/mL of ethidium bromide, 3 V/cm).

### 2.3. Evaluating the Effect of Pollenkitt on PCR

To test our hypothesis of pollenkitt inhibition of PCR, we conducted a series of PCR experiments with different dilutions of pollenkitt, using genomic DNA of *A. cepa* as a template and the LFS gene as an amplification target. PCR mix, cycling conditions, and gel electrophoresis were performed as in the previous section (positive control). Pollenkitt was extracted from pollen grains as the following: in a 1.5 mL tube, 1 mg pollen was thoroughly vortexed with 100 μL $ddH_2O$ for 30 s then centrifuged for 5 s using minispin centrifuge ($750\times g$). Since pollen grains are much heavier than the pollenkitt debris, they are precipitated first, and the latter remains in the supernatant. 1 μL of the supernatant was added to the PCR reaction either immediately (concentrated) or as an aqueous dilution (1:10 and 1:100).

### 2.4. Microscopy of the Onion Pollenkitt

For microscope preparation, a large droplet of distilled water (100 μL) was placed on a glass slide. Using a microspatula or dissecting needle, clumps of pollen (~1 mg) were carefully placed on the droplet. Microscopy was performed using Zeiss SterREO LUMAR.V12 and Zeiss Axiolab 5.

### 2.5. Wet Manual Pollination of Onion Flowers

Two different liquid mediums were tested in wet pollination experiments: (1) 12% (*w/v*) sucrose water solution and (2) pollen growing medium according to Brewbaker and Kwack [30] as following: 0.1 g/L $H_3BO_3$, 0.3 g/L $Ca(NO_3)_2\cdot4H_2O$, 0.3 g/L $MgSO_4\cdot7H_2O$, 0.1 g/L $KNO_3$ dissolved in 12% sucrose. The pollen clump was added to 100 μL of liquid medium placed on a glass slide. The floating cover of pollen was easily transferred to the stigma using a small bacterial inoculation loop.

About 70–75% of the flowers in selected umbels have been removed to make subsequent steps more convenient. Umbels were proven against flying insects using isolation cages made of 4-layered medical gauze. Castration was conducted on a daily basis. Due to the non-synchronous flowering, the umbels were constantly checked to immediately remove the immature anthers from the newly opened flowers. The flowering period for a single umbel (the time between the opening of the first flower until the drying of the last flower) lasted for 8–10 days on average.

### 3. Results

During experimental studies on the genetic manipulation of onion pollen, we encountered a number of hidden pitfalls due to the lack of a basic methodology for evaluating

results and post-experimental application. To address the lack of practical knowledge, we investigated the factors that influence the polymerase chain reaction (PCR) on pollen grains, the viability of pollen, and its further use for pollination and seed setting.

### 3.1. Direct PCR on Pollen Suspension of Onion

A gene encoding Lachrymatory Factor Synthase (LFS) was used as a target for PCR on pollen. This is a single-locus gene encoding enzyme involved in the synthesis of the volatile sulfur compounds released during wounding [31,32]. In the first attempt of PCR using fresh intact pollen as a template we faced a complete inhibition of amplification. During the early stages of our experiment, we believed that the inaccessibility of pollen DNA could be the reason for unsuccessful PCR. Most of the available protocols for PCR on pollen require DNA extraction [10,12,18,33]. Accordingly, we tried different techniques of DNA extraction through mechanical damage of the pollen wall: vortexing, grinding with a pestle, steel beads, or glass powder. None of these techniques showed any positive PCR results (data not shown). Finally, a simple washing procedure using distilled water was able to restore normal amplification (Figure 1). We concluded that an unknown factor/substance coating the surface of the pollen grain might be responsible for PCR inhibition.

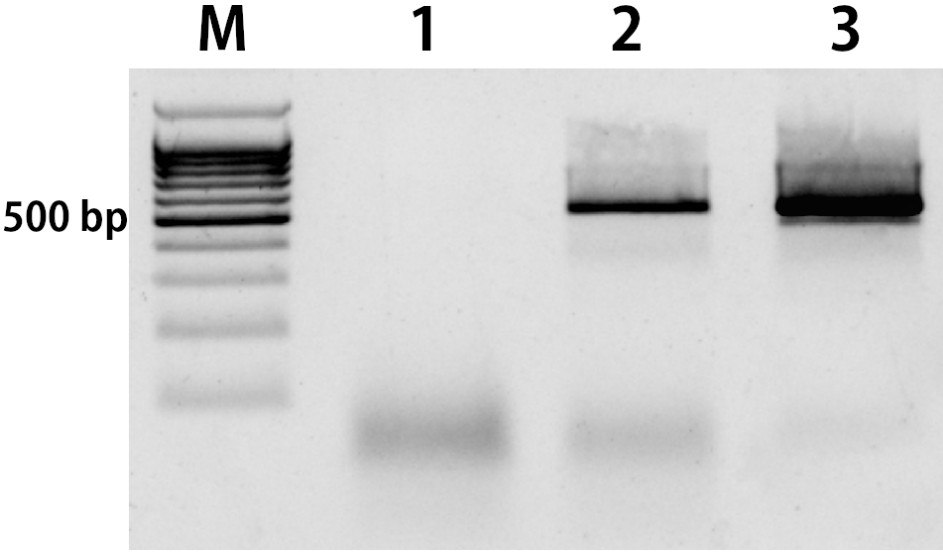

**Figure 1.** PCR amplification of the gene encoding LFS in onion: M—DNA Ladder 100bp+; lane 1—mature pollen without washing; lane 2—mature pollen after 10-times washing in ddH$_2$O; lane 3—genomic DNA of *A. cepa*.

In all angiosperms, the common material for coating pollen grains is the sticky tapetal tissue secretion, also known as pollenkitt [34]. In addition to a wide range of functions [26], pollen serves as a binding agent in entomophilous plants [35]. According to Dobson [36], pollenkitt is a hydrophobic mixture of materials composed mainly of saturated and unsaturated lipids. Lipids are known to be major inhibitors of PCR [37].

During the washing procedure of the pollen suspension, we noticed that the discarded supernatant always had a cloudy appearance. Since pollenkitt is a hydrophobic substance, we assumed that the cloudiness is probably due to the insoluble debris of the pollenkitt, which is released into the water during washing (vortexing and centrifugation). This was confirmed using dark field microscopy (Figure 2a,b).

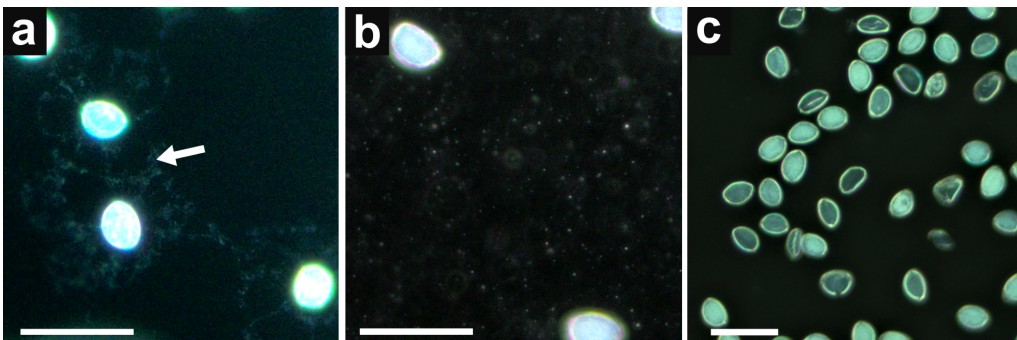

**Figure 2.** The dark field microscopy of pollen suspension. (**a**) pollen before washing—pollenkitt appears as a bright material around each pollen grain. White arrow pointing to the released pollenkitt from pollen grains; (**b**) supernatant after vortexing and centrifugation—pollenkitt is no longer accumulated around each pollen grain, but rather scattered into fine granular particles (appears here as bright white dots); (**c**) pollen suspension after washing—the pollenkitt is removed. Bars represent 50 μm.

Upon repeated washing of the suspension, pollenkitt residues were removed together with the supernatant until their concentration became negligible for PCR inhibition (Figure 2c). For onion pollen, 10-times washing by vortexing in 500 μL of distilled water for 10 s and centrifuging for 5 s using minispin centrifuge (750× *g*) was optimal.

To verify the relationship between PCR inhibition and pollenkitt, we conducted a series of PCR with different dilutions of pollenkitt added to genomic DNA of *A. cepa* as an amplification template and the LFS gene as a target (see Materials and Methods). Electrophoresis showed a clear correlation between pollenkitt concentration and the intensity of the PCR band (Figure 3).

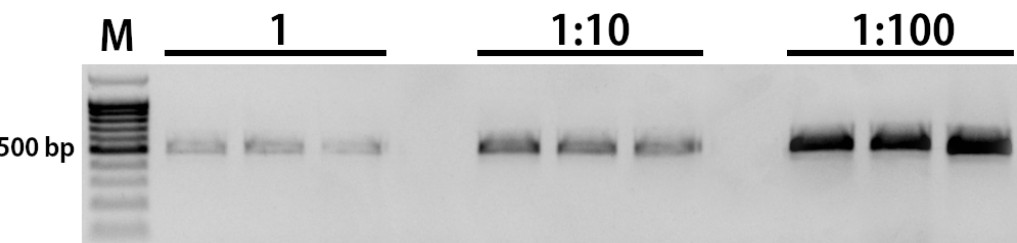

**Figure 3.** Inhibition of PCR upon addition of aqueous dilutions of pollenkitt to *A. cepa* genomic DNA as an amplification template and the LFS gene as a target. The PCR experiment was repeated three times with each dilution. Dilution 1 represents the equivalent amount of pollenkitt eluated from 10 μg of pollen.

### 3.2. Hand Pollination with Wetted Pollen Grains

Most existing methods of delivery of the editing systems into pollen are carried out in a wet medium. One of the main objectives of our study was to verify whether it is possible to pollinate onion flowers using wetted pollen grains. Since hydration is an inevitable step in every technique of pollen genetic manipulation, it is crucial to verify such matters. In vitro experiments of pollen tube growing we found that onion pollen grains cannot grow after dehydration. Thus, drying the suspension was excluded. The necessity of the use of wet pollen led us to conduct detailed studies of the dynamics of pollen wetting using microscopy.

Imaging under a stereo microscope showed a rapid movement of the pollen clump immediately upon contact with liquid. This rapid movement was mainly caused by the momentary swelling of the pollen grains (Video S1). Despite the vigorous movement, it was noticeable that the neighboring pollen grains could not come into direct contact with each other, as if they were surrounded by an invisible structure (Video S2). In less than a minute, the rapid movement stopped, and all the pollen grains that were previously in

one clump floated steadily on the surface of the droplet. Using phase-contrast microscopy a light-dense structure surrounding the floating pollen grains was observed (Figure 4a). We suggest that this structure is in fact the pollenkitt released from the surface of pollen grains upon contact with water. Within 4–5 min after contact with the liquid surface, adjacent fragments of pollenkitt began to join together, forming an interwoven "blanket" of pollenkitt components (Figure 4b).

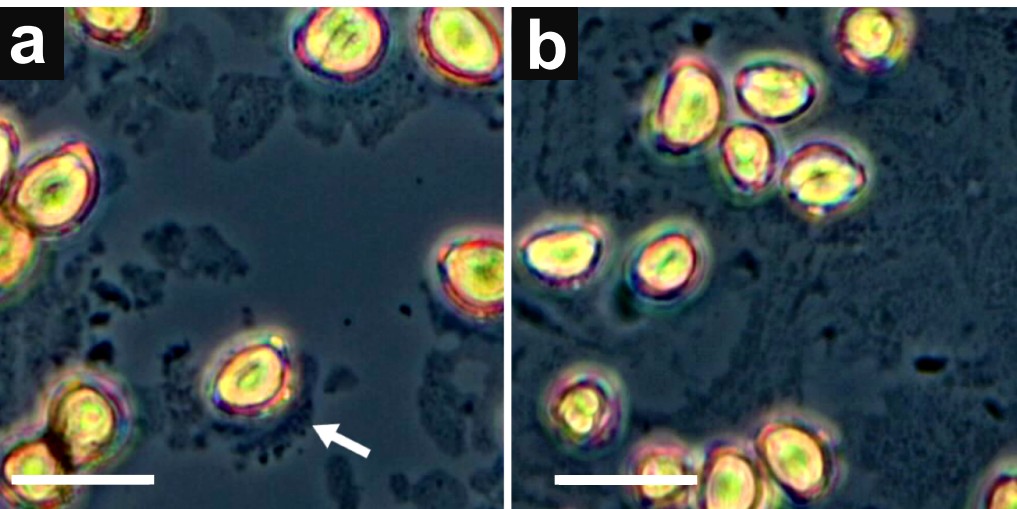

**Figure 4.** The phase-contrast microscopy of the pollenkitt in *A. cepa*: (**a**) pollen grains immediately after placing on a droplet of distilled water; pollenkitt appears as fragments released from the surface of pollen grains (white arrow). (**b**) the same pollen grains after 5 min; pollenkitt appears as a "blanket" that keeps pollen grains on the surface of the droplet. Bars represent 50 μm.

This "blanket" obviously affects the floating and adhesion properties of pollen grains. Based on this observation the optimal technique of transferring the wet pollen grains onto the stigma was developed. Using an inoculation loop, a lump of the blanket was easily collected from a liquid surface by the hollow center of the loop which enabled us to easily apply an even layer of wetted pollen grains to the stigmatic surface (Video S3).

Two different liquid mediums were tested: (1) 12% (*w/v*) sucrose water solution and (2) pollen growing medium according to Brewbaker and Kwack [30] as following: 0.1 g/L $H_3BO_3$, 0.3 g/L $Ca(NO_3)_2 \cdot 4H_2O$, 0.3 g/L $MgSO_4 \cdot 7H_2O$, 0.1 g/L $KNO_3$ dissolved in 12% sucrose. There were two positive controls: (3) open pollination with insects and (4) hand pollination with fresh intact pollen grains using a brush. A negative control (5) was also used to test castration and inflorescence isolation.

The seed set percentage using pollen growing medium was the lowest—35.7%, while open pollination by insects was the highest—93.5% (Table 1). The percentage of set seeds in the experiment with 12% sucrose water solution was almost the same as in the positive control of intact pollen grains (63.5% and 65.5%, respectively). This result means that 12% sucrose water solution and the method of applying pollen using a loop are the most optimal for further work related to pollen genetic manipulations.

**Table 1.** The effect of different pollination methods on seed setting in the *A. cepa* umbels.

| Treatment | Number of Flowers in Umbel | Number Flowers that Set Seeds, (%) |
| --- | --- | --- |
| Wet pollination with pollen growing medium | 56 | 20 (35.7) |
| Wet pollination with 12% sucrose solution | 52 | 33 (63.5) |
| Manual pollination with un-wetted pollen grains | 61 | 40 (65.5) |
| Open pollination by insects | 62 | 58 (93.5) |
| Negative control | 0 | 0 (0) |

In summary, our results demonstrate a feasible technique of using wetted pollen grains to pollinate onion flowers without affecting viability and seed setting. Minimizing the influence of technical procedures is always favorable, especially in experiments such as genetic manipulation. Therefore, engaging our wet pollination technique, and PCR using pollen DNA as a template might be very beneficial in future experiments based on genetic manipulation of onion pollen aimed at producing seeds with an edited genome.

**Supplementary Materials:** The following supporting information can be downloaded at: https://www.mdpi.com/article/10.3390/cimb45020070/s1, Video S1: The wetting behavior of onion pollen after contact with liquid. The movie was recorded in real-time using Zeiss SterREO LUMAR.V12 (objective: NeoLumer S 0.8x FWD 80mm); Video S2: Pollenkitt release from onion pollen grains after contact with liquid. The movie was recorded in real-time using Zeiss SterREO LUMAR.V12 (objective: NeoLumer S 0.8x FWD 80 mm). The fixed image is the same area shown under phase-contrast microscope Zeiss Axiolab 5; Video S3: The technique we used for manual pollination of onion flower shown.

**Author Contributions:** M.M.—investigation, validation, visualization, writing—original draft, methodology; A.E.—investigation, validation, writing—original draft, methodology, L.K. — conceptualization, funding acquisition, project administration, supervision, writing—original draft, writing—review & editing. All authors have read and agreed to the published version of the manuscript.

**Funding:** This research was funded by the Russian Foundation for Basic Research, grant number 20-016-00065.

**Institutional Review Board Statement:** Not applicable.

**Informed Consent Statement:** Not applicable.

**Data Availability Statement:** Data available in a publicly accessible repository.

**Acknowledgments:** We thank Alexander Voronkov, K.A. Timiryazev Institute of Plant Physiology RAS, for practical advice on pollen collection.

**Conflicts of Interest:** The authors declare no conflict of interest.

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
