# Peer review of "Hidden Pitfalls of Using Onion Pollen in Molecular Research"

_cimb, doi:10.3390/cimb45020070_

Round 1
Reviewer 1 Report
General Comments: The topic is interesting and valuable regarding the gene editing and manipulation in the field of molecular biology. However authors need to well-structured the manuscript. For that the manuscript needs to improve mostly the materials and methods section by explaining clearly the necessity and impact of wet pollination in this context. The images (Figure 2 and 4) are not clear enough to detection of pollenkitt. The images are in very low magnification and resolution that does not visibly distinguish the exine and intine. Improve the image quality that clearly focus on pollenkitt and visualize distinctly.
Line No Remarks
15 Arrange alphabetically
46-47 Is there any significance to writing in italicized form?
112 "x" is not used traditionally. Use "."
170 Written as stereo-microscope
189 Follow the earlier comment of line no – 112
Reviewer 2 Report
Review Report
Manuscript ID: cimb-2183504
The manuscript entitled “Hidden Pitfalls of Using Onion Pollen in Molecular Research” submitted to CIMB needs corrections and improvements. Generally, the manuscript is fine but still there are some minor comments for the improvement of the manuscript.
Abstract: In the abstract part please mention the specific objective of your study
Introduction: In the introduction part please mention the history, importance and background of onion pollen and their use in molecular research
Materials and Methods: Line 64 to 69 please mentions the date of planting, flowering and maturity of onion. Please also mention the site of the experiment. Line 71 mention the manufacturer name regarding analytical balance and line 70 to 92 reference missing regarding DNA extraction and PCR Analysis.
Carefully check all spellings, numerical order of references, format following the requirements of the submitted journals.
Cross check all the references cited in the text with reference list
Reviewer 3 Report
Please, see an enclosed file.
